# GAUSSIANFUSIONOCC: A SEAMLESS SENSOR FUSION APPROACH FOR 3D OCCUPANCY PREDICTION USING 3D GAUSSIANS

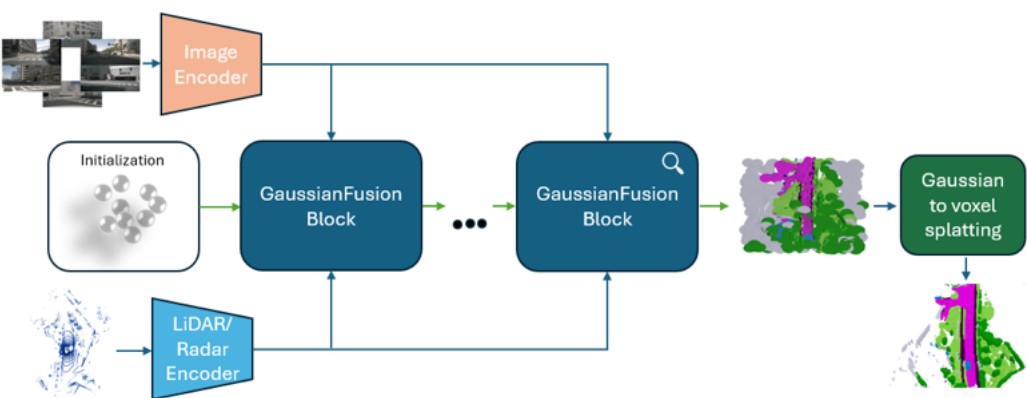

Figure 1: **GaussianFusionOcc pipeline:** Our method utilizes modality-specific encoders to extract feature maps from the input data. Extracted features are then fed to GaussianFusion blocks along with Gaussians and queries from the previous block. The GaussianFusion block, visualized in Figure 2, extracts per-Gaussian features from each modality, fuses them into a unified feature vector, which is then used to refine the input Gaussians. Resulting Gaussians from the last block are splatted to a voxelized representation using the Gaussian-to-voxel splatting module.

## ABSTRACT

3D semantic occupancy prediction is one of the crucial tasks of autonomous driving. It enables precise and safe interpretation and navigation in complex environments. Reliable predictions rely on effective sensor fusion, as different modalities can contain complementary information. Unlike conventional methods that depend on dense grid representations, our approach, GaussianFusionOcc, uses semantic 3D Gaussians alongside an innovative sensor fusion mechanism. Seamless integration of data from camera, LiDAR, and radar sensors enables more precise and scalable occupancy prediction, while 3D Gaussian representation significantly improves memory efficiency and inference speed. GaussianFusionOcc employs modality-agnostic deformable attention to extract essential features from each sensor type, which are then used to refine Gaussian properties, resulting in a more accurate representation of the environment. Extensive testing with various sensor combinations demonstrates the versatility of our approach. By leveraging the robustness of multi-modal fusion and the efficiency of Gaussian representation, GaussianFusionOcc outperforms current state-of-the-art models.

## 1 INTRODUCTION

Accurate 3D semantic occupancy prediction is a foundational task for safe autonomous navigation. Reliable perception of the surrounding environment enables precise situational awareness necessary

for informed decision-making, efficient route planning, and collision avoidance (Ming et al., 2024; Lu et al., 2024; Caesar et al., 2020; Wang et al., 2023). While recent advances in voxel-based methods (Li et al., 2023a; Jiang et al., 2024; Miao et al., 2023; Wei et al., 2023; Zhang et al., 2023) have pushed performance on benchmarks like nuScenes (Caesar et al., 2020; Wei et al., 2023; Wang et al., 2023), their reliance on dense volumetric grids creates prohibitive computational and memory costs (Huang et al., 2023; 2024b;a). Furthermore, these methods struggle with the inherent sparsity of relevant information in driving scenes, where most of the volume remains empty. Concurrently, sensor fusion remains critical for robustness and reliability in dynamic environments, as cameras, LiDAR, and radar provide complementary strengths: cameras capture rich semantics, LiDAR offers precise geometry, and radar ensures reliability in adverse conditions. However, existing fusion frameworks are constrained by their underlying representations, which lack adaptability to scene complexity and scalability across modalities Liu et al. (2023); Ming et al. (2024); Wang et al. (2023).

Emerging 3D Gaussian splatting techniques (Kerbl et al., 2023; Huang et al., 2024b;a, Yang et al., 2024) offer a promising alternative to voxels by modeling scenes with anisotropic, learnable Gaussians, enabling faster rendering and lower memory usage. The inherent sparsity and adaptability of Gaussian representations naturally align with the characteristics of driving scenes, where information density varies across the environment (Huang et al., 2024b). By concentrating representational capacity where it matters most, Gaussian-based methods achieve superior memory efficiency.

We present GaussianFusionOcc, a novel framework that extends the capabilities of Gaussian-based 3D scene representations to the multi-modal setting. While existing Gaussian-based approaches (Huang et al., 2024b;a) have been limited to single-modality inputs, our approach introduces a modality-agnostic Gaussian encoder, capable of extracting per-Gaussian features from any individual sensor modality, such as camera, LiDAR, or radar, using a deformable attention mechanism. This attention-based approach enables the encoder to focus computational resources on the most informative regions of each sensor's feature map, ensuring robust performance even when certain sensors are impaired or provide conflicting information. Unlike traditional multi-modal fusion approaches that aggregate features at fixed spatial locations with uniform resolution, we propose a novel fusion paradigm that constructs unified Gaussian feature vectors from modality-specific features, fusing information directly on learnable geometric-semantic primitives. This shift enables the model to adaptively allocate Gaussians based on scene complexity, focusing modeling capacity on detailed or semantically rich regions while maintaining sparsity elsewhere.

The main contributions of this work are the following:

We introduce the first framework that leverages 3D Gaussian splatting for multi-modal 3D semantic occupancy prediction and fuses the sensor features in Gaussian space.

We propose a modality-agnostic Gaussian encoder based on a deformable attention mechanism that effectively extracts per-Gaussian features from diverse sensor modalities, and a fusion method that creates a unified representation based on the extracted features.

We demonstrate state-of-the-art performance on the nuScenes dataset, especially on the rainy and nighttime subsets, achieving superior performance accuracy while reducing memory requirements compared to leading voxel-based approaches.

We tested the model with various sensor combinations under different scenarios to demonstrate the performance gains.

## 2 RELATED WORK

Earlier 3D semantic occupancy prediction approaches (Li et al., 2023a; Cao & De Charette, 2022; Jiang et al., 2024; Miao et al., 2023; Wei et al., 2023; Zhang et al., 2023) mostly relied on dense voxel grids to represent the 3D scene. While effective in capturing fine-grained details, these dense grid-based methods often suffer from high computational and memory overhead due to the inherent sparsity of real-world environments and the need for high-resolution grids. To mitigate these limitations, recent approaches Li et al. (2024); Huang et al. (2023); Li et al. (2023b) explored alternative scene representations like Bird's-Eye-View (BEV) and Tri-Perspective View (TPV), achieving strong performance with improved efficiency. However, these planar representations often

involve a compression of 3D information, potentially leading to a loss of fine-grained geometric details necessary for accurate 3D occupancy prediction.

In pursuit of more efficient and scalable 3D scene representations, object-centric approaches (Huang et al., 2024b;a; Gan et al., 2024; Yang et al., 2024; Lu et al., 2024; Wang et al., 2024) have emerged as efficient alternatives to dense voxel grids, representing scenes using primitives centered around objects or regions of interest to avoid computational redundancy. Among these, 3D Semantic Gaussians have recently gained traction as a flexible and efficient representation for 3D scenes (Huang et al., 2024b;a; Gan et al., 2024; Yang et al., 2024), with each Gaussian primitive capturing local geometric and semantic information. GaussianFormer (Huang et al., 2024b) introduced an object-centric approach using 3D semantic Gaussians for vision-based 3D semantic occupancy prediction, demonstrating comparable performance to state-of-the-art methods with significantly reduced memory consumption. By representing the scene with a set of learnable 3D Gaussians and employing efficient Gaussian-to-voxel splatting for occupancy prediction, GaussianFormer showcases the potential of object-centric representations for achieving both accuracy and efficiency. Probabilistic extensions (Huang et al., 2024a) further aim to improve the utilization and efficiency of 3D Gaussian representations.

Recognizing the limitations of relying solely on a single sensor, multi-sensor fusion has become a critical direction for robust 3D occupancy prediction. Integrating LiDAR data with camera images has been shown to improve depth estimation and overall perception accuracy. BEVFusion (Liu et al., 2023) proposed fusing LiDAR and camera features in the BEV space for multi-task perception. SparseFusion (Xie et al., 2023) further refined the feature fusion module for improved efficiency. While these methods primarily focus on 3D object detection (Liu et al., 2023; Xie et al., 2023; Chen et al., 2023), there's a growing interest in multi-sensor fusion for 3D semantic occupancy prediction. Camera-radar fusion has also been explored for tasks like object detection and tracking (Kim et al., 2023; Nabati & Qi, 2021; Chen et al., 2023), but dedicated approaches for 3D semantic occupancy prediction are scarce due to the sparsity of radar data. To address the need for robust occupancy prediction, OccFusion (Ming et al., 2024) was introduced as a novel framework to integrate features from surround-view cameras, radars, and 360-degree LiDAR through dynamic fusion modules, demonstrating superior performance in challenging conditions. These efforts highlight the benefits of combining complementary sensor data to achieve more reliable and accurate 3D scene understanding for autonomous driving. Building upon these advancements, our GaussianFusionOcc leverages the efficiency and flexibility of 3D semantic Gaussians while introducing a seamless sensor fusion mechanism to harness the complementary strengths of camera, LiDAR, and radar data for robust and accurate 3D occupancy prediction.

## 3 METHODS

In this section, we present our Gaussian-space sensor fusion approach for 3D semantic occupancy prediction, which fuses information directly on learnable 3D Gaussian primitives. This enables adaptive spatial allocation that scales sublinearly with respect to the scene resolution. Our approach extracts features from sensors using sensor-specific encoders. Sensor-specific features are then fed into the Gaussian encoder blocks, where modality-agnostic deformable attention (Zhu et al., 2020) is utilized to extract Gaussian-centric features. These features are later fused at the Gaussian level and used to iteratively refine the properties of probabilistic semantic 3D Gaussians, which act as persistent carriers of multi-modal evidence. Finally, these refined Gaussians are splatted to generate a 3D voxel grid representation.

**Sensor feature extraction:** We employ tailored feature extraction pipelines for each modality to effectively encode information from our diverse set of sensors.

For the surrounding camera images $I = \{I_i \in \mathbb{R}^{3 \times H \times W}\}_{i=1}^{N}$ where H, W, N are height, width, and number of cameras, we utilize a combination of ResNet101-DCN (He et al., 2016; Dai et al., 2017) as our 2D backbone and a Feature Pyramid Network (FPN) (Lin et al., 2017) as the neck. This architecture allows us to generate multi-scale image features $F_i^{cam} = \{F_{i,j}^{cam} \in \mathbb{R}^{C_i \times H_i \times W_i}\}_{j=1}^{M}$ for $i$-th image where M is the number of scales.

To process the LiDAR point cloud $P^{lidar} \in \mathbb{R}^{C \times H \times W}$, we first perform voxelization of the input data. Following this, we employ a VoxelNet (Zhou & Tuzel, 2018) encoder along with an FPN (Lin et al., 2017) to produce multi-scale Bird's-Eye-View (BEV) features $F^{lidar} = \{F_i^{lidar} \in \mathbb{R}^{C_i \times H_i \times W_i}\}_{i=1}^{M}$.

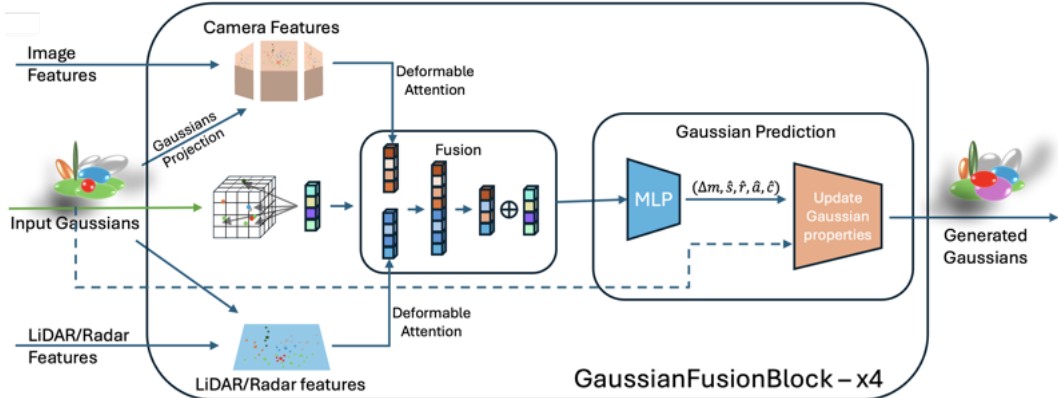

Figure 2: **GaussianFusionBlock architecture.** Modality-agnostic Gaussian encoders employ a deformable attention mechanism to extract relevant features for each initial Gaussian. These per-Gaussian features are then fed to the fusion module along with the resulting features of the sparse convolution layer applied to voxelized Gaussians. Fusion module concatenates Gaussian encoder features, applies an MLP fuser, and adds sparse convolution features to create unified Gaussian features. These features are then used in the Gaussian prediction module to refine initial Gaussians by predicting mean offset, scale, rotation, opacity, and semantic class.

Our radar encoder first pillarizes the input point cloud $P^{radar} \in \mathbb{R}^{C \times H \times W}$. The pillarization is a form of voxelization where the 3D space is divided into vertical pillars. It then encodes the radar features using a PointPillars (Lang et al., 2019) voxel encoder and a middle encoder. Radar data, while providing valuable information about object velocity, typically yields sparse features $F^{radar} \in \mathbb{R}^{C \times H \times W}$ compared to camera and LiDAR.

Inspired by the advancements in GaussianFormer (Huang et al., 2024b) and GaussianFormer-2 (Huang et al., 2024a), our method reconstructs semantic probabilistic 3D Gaussians to represent the scene occupancy. This object-centric approach aims to overcome the limitations of dense grid-based representations by sparsely modeling the scene with learnable Gaussians. To initialize the Gaussian properties, we randomly sample a desired number of Gaussians $\mathcal{G} = \{m_i, s_i, r_i, a_i, c_i\}_{i=1}^{P}$, use learnable parameters for initialization, or choose the desired number of points from the input LiDAR point cloud for Gaussian means. Our model comprises three key modules: Gaussian encoder, fusion module, and Gaussian prediction module.

**Gaussian encoder**: To refine initial Gaussians, our method uses independent instances of the modality-agnostic Gaussian encoder module, which extracts a feature vector for each Gaussian. The encoder samples multiple 3D reference points around the center of a Gaussian $m$ while taking into consideration the Gaussian's shape and size determined by its scale and rotation matrices $S, R$:

$$\Delta m = RS\Phi^{offset}(q), \quad R = \{m + \Delta m_i\}_{i=1}^{N_R} \tag{1}$$

where $\Phi^{offset}, q$ denote the MLP for offset prediction, and the input query for the specific Gaussian, respectively. Initial queries for the first block are randomly sampled. These 3D reference points are then projected onto the feature maps of the input sensors using the sensors' intrinsic and extrinsic parameters. This projection establishes a correspondence between the Gaussian and the 2D features captured by the sensors. To extract relevant information from the sensor feature maps at the projected reference points, the Gaussian Encoder utilizes a deformable attention function (Zhu et al., 2020):

$$F^{GE} = \sum_{i=1}^{N} \sum_{j=1}^{N_R} DA(Q, P(R), F^{sensor}) \tag{2}$$

where $DA(), P(), F^{sensor}, N, N_R$ denote deformable attention function, projection from world to sensor features coordinates, sensor feature maps, number of sensor inputs (number of cameras or radars), and number of reference points, respectively. The attention mechanism computes a weighted sum of the features sampled around these refined reference points, where the weights indicate the

relevance of each sampled feature. This aggregated feature vector represents the visual (or other sensory) information associated with the 3D Gaussian.

**Fusion:** The fusion module is designed to effectively integrate multi-modal sensor information to generate a unified feature representation for each 3D Gaussian. Given a set of per-Gaussian feature vectors extracted from individual sensors such as cameras, LiDAR, and radar, the fusion module aims to create a comprehensive descriptor that leverages the complementary strengths of each modality. The feature vectors derived from each available sensor modality are concatenated along the feature channel dimension. This operation results in a single, extended feature vector that aggregates information from all input sensors. If we denote the feature vector for a Gaussian from the $i$-th sensor as $F_i$, the concatenation operation can be represented as: $F^{concatenated} = [F_1, F_2, ..., F_n]$ where $n$ is the number of sensor modalities being utilized. Following the concatenation, the combined feature vector is passed through a Multi-Layer Perceptron $\Phi^{fusion}$. The MLP serves to learn complex interdependencies between features from different sensors, weigh the contribution of different sensor features based on their relevance for representing the 3D Gaussian, and reduce the dimensionality of the concatenated feature vector to a more manageable and informative unified representation.

The feature vector for each 3D Gaussian on the output of the MLP encapsulates the integrated information from all sensor modalities, providing a robust descriptor of the local 3D space. In parallel, input Gaussians are represented as a point cloud of their means. The voxelized representation is then fed to 3D sparse convolutions to leverage the spatial relationships between these Gaussians, inspired by the self-encoding module from GaussianFormer (Huang et al., 2024b). Sparse convolutions $SC$ are particularly suitable for processing sparse 3D data as they only operate on occupied voxels, allowing each Gaussian to gather contextual information from neighboring Gaussians. Finally, to combine the benefits of both pathways, the sparse convolution outputs are added to the MLP-generated feature vectors for each corresponding Gaussian:

$$Q = F^{unified} = \Phi^{fusion}(F^{concatenated}) \oplus SC(m) \tag{3}$$

This element-wise addition fuses the modality-specific information (refined by the MLP) with the spatial context information (learned by the sparse convolutions). These unified Gaussian feature vectors are subsequently used for refining the properties of the Gaussians, ultimately contributing to the accurate prediction of 3D semantic occupancy.

**Gaussian prediction:** Leveraging the information aggregated from multi-modal sensor inputs, the Gaussian Prediction module updates the parameters of each Gaussian to better represent the surrounding 3D scene. Each Gaussian acts as a persistent carrier of multi-modal evidence across refinement blocks. The functionality of the Gaussian prediction module draws inspiration from the refinement module employed in GaussianFormer (Huang et al., 2024b). Specifically, for each 3D Gaussian, the unified feature vector from the Fusion module serves as input to a multi-layer perceptron $\Phi^{refine}$. This MLP decodes intermediate Gaussian properties, including offset to the mean ($\hat{m}$), scale ($\hat{s}$), rotation ($\hat{r}$), opacity ($\hat{a}$), and semantic logits ($\hat{c}$):

$$(\hat{m}, \hat{s}, \hat{r}, \hat{a}, \hat{c}) = \Phi^{refine}(Q) \tag{4}$$

Following the refinement step, the updated Gaussian properties and unified Gaussian feature vectors are fed to the next Gaussian fusion block for subsequent refinement. This iterative refinement process allows for a progressively more accurate and semantically meaningful representation of the 3D scene. After the final refinement, the collection of refined 3D Gaussians is used to generate the final 3D semantic occupancy prediction. To achieve this, we employ the Gaussian-to-Voxel splatting method $GS$ proposed in GaussianFormer-2 (Huang et al., 2024a):

$$o(x) = GS(m, s, r, a, c) \tag{5}$$

where x denotes the voxel position. This approach utilizes probabilistic Gaussian superposition to transform the sparse Gaussian representation into a dense voxel grid occupancy prediction.

To train GaussianFusionOcc, we use lovasz-softmax loss (Berman et al., 2018) $L_{lovasz}$ and binary cross entropy loss $L_{BCE}$. During the training, the model splats Gaussians to a voxel grid representation in each iteration of the refinement, so the losses could be calculated for each iteration. If $N$ denotes the number of refinement iterations, we can then express the total loss as:

$$L^{total} = \sum_{i=1}^{N}(L_i^{lovasz} + L_i^{BCE}) \tag{6}$$

# 4 EXPERIMENTS

In this section, we present a comprehensive evaluation of our proposed GaussianFusionOcc framework for 3D semantic occupancy prediction. We assess its performance on the nuScenes validation set, including challenging rainy and nighttime scenarios, and analyze the contribution of key components through ablation studies. Efficiency comparisons are also provided.

## 4.1 DATASET DETAILS

We use the nuScenes dataset (Caesar et al., 2020) for training and evaluation, which contains 1000 scenes with keyframes annotated at 2Hz, each equipped with 6 surround-view cameras, 5 radars, and 1 LiDAR. Since nuScenes lacks native dense 3D semantic occupancy ground truth, we use annotations generated by SurroundOcc (Wei et al., 2023). The occupancy prediction range spans [-50m, 50m] for X and Y axes and [-5m, 3m] for the Z axis, with 0.5-meter voxel resolution. We trained the model on the official training split (700 scenes) and evaluated on the validation split (150 scenes).

## 4.2 IMPLEMENTATION DETAILS

GaussianFusionOcc uses a ResNet101-DCN (He et al., 2016; Dai et al., 2017) as the image feature backbone, initialized with weights from the FCOS3D (Wang et al., 2021) checkpoint. Resulting image feature maps are then processed using FPN (Lin et al., 2017) to create multi-scale image features with 1/4, 1/8, 1/16, and 1/32 of the original resolution. For the LiDAR encoder, it uses VoxelNet (Zhou & Tuzel, 2018) with weights from the FUTR3D (Chen et al., 2023) checkpoint, and FPN (Lin et al., 2017) producing multi-scale image features with the same downsampling as for image features. Radar encoder uses voxel encoder and middle encoder from PointPillars (Lang et al., 2019) network initialized with weights from FUTR3D (Chen et al., 2023) checkpoint. For the 3D Gaussian representation, we use 6400 Gaussians in our experiments. The model consists of 4 Gaussian fusion blocks, each consisting of Gaussian encoders, a fusion module, and a Gaussian prediction module. The Gaussian encoder and fusion module produce features with 128 channels. We train our model for 20 epochs with a batch size of 8 on an NVIDIA RTX A6000 GPU. We employ the AdamW optimizer (Loshchilov & Hutter, 2017) with a weight decay of 0.01. The learning rate is warmed up to 2e-4 in the first 500 iterations, and subsequently decreased following a cosine schedule. For data augmentation, we apply photometric distortions to the input images.

## 4.3 MAIN RESULTS

Table 1: 3D semantic occupancy prediction results on nuScenes validation set. * marks the model supervised by dense occupancy annotations, while the original was trained with LiDAR segmentation labels. ** marks the model using LiDAR initialization, *** the random initialization, while other models use learnable initialization. Modality notation: Camera (C), LiDAR (L), Radar (R)

| Method | Modality | IoU | mIoU | barrier | bicycle | bus | car | constr. veh. | motorcycle | pedestrian | traffic cone | trailer | truck | driveable surf. | other flat | sidewalk | terrain | man made | vegetation |
|---|---|---|---|---|---|---|---|---|---|---|---|---|---|---|---|---|---|---|---|
| MonoScene (Cao & De Charette, 2022) | C | 23.96 | 7.31 | 4.03 | 0.35 | 8.00 | 8.04 | 2.90 | 0.28 | 1.16 | 0.67 | 4.01 | 4.35 | 27.72 | 5.20 | 15.13 | 11.29 | 9.03 | 14.86 |
| Atlas (Murez et al., 2020) | C | 28.66 | 15.00 | 10.64 | 5.68 | 19.66 | 24.94 | 8.90 | 8.84 | 6.47 | 3.28 | 10.42 | 16.21 | 34.86 | 15.46 | 21.89 | 20.95 | 11.21 | 20.54 |
| BEVFormer (Li et al., 2024) | C | 30.50 | 16.75 | 14.22 | 6.58 | 23.46 | 28.28 | 8.66 | 10.77 | 6.64 | 4.05 | 11.20 | 17.78 | 37.28 | 18.00 | 22.88 | 22.17 | 13.80 | 22.21 |
| TPVFormer (Huang et al., 2023) | C | 11.51 | 11.66 | 16.14 | 7.17 | 22.63 | 17.13 | 8.83 | 11.39 | 10.46 | 8.23 | 9.43 | 17.02 | 8.07 | 13.64 | 13.85 | 10.34 | 4.90 | 7.37 |
| TPVFormer* (Huang et al., 2023) | C | 30.86 | 17.10 | 15.96 | 5.31 | 23.86 | 27.32 | 9.79 | 8.74 | 7.09 | 5.20 | 10.97 | 19.22 | 38.87 | 21.25 | 24.26 | 23.15 | 11.73 | 20.81 |
| OccFormer (Zhang et al., 2023) | C | 31.39 | 19.03 | 18.65 | 10.41 | 23.92 | 30.29 | 10.31 | 14.19 | 13.59 | 10.13 | 12.49 | 20.77 | 38.78 | 19.79 | 24.19 | 22.21 | 13.48 | 21.35 |
| SurroundOcc (Wei et al., 2023) | C | 31.49 | 20.30 | 20.59 | 11.68 | 28.06 | 30.86 | 10.70 | 15.14 | 14.09 | 12.06 | 14.38 | 22.26 | 37.29 | 23.70 | 24.49 | 22.77 | 14.89 | 21.86 |
| GaussianFormer (Huang et al., 2024b) | C | 29.83 | 19.10 | 19.52 | 11.26 | 26.11 | 29.78 | 10.47 | 13.83 | 12.58 | 8.67 | 12.74 | 21.57 | 39.63 | 23.28 | 24.46 | 22.99 | 9.59 | 19.12 |
| GaussianFormer-2 (Huang et al., 2024a) | C | 31.74 | 20.82 | 21.39 | 13.44 | 28.49 | 30.82 | 10.92 | 15.84 | 13.55 | 14.04 | 22.92 | 40.61 | 24.36 | 26.08 | 24.27 | 13.83 | 21.98 |
| L-CONet (Wang et al., 2023) | L | 39.40 | 17.70 | 19.20 | 4.00 | 15.10 | 26.90 | 6.20 | 3.80 | 6.80 | 6.00 | 14.10 | 13.10 | 39.70 | 19.10 | 24.00 | 23.90 | 25.10 | 35.70 |
| M-CONet (Wang et al., 2023) | C+L | 39.20 | 24.70 | 24.80 | 13.00 | 31.60 | 34.80 | 14.60 | 18.00 | 20.00 | 14.70 | 20.00 | 26.60 | 39.20 | 22.80 | 26.10 | 26.00 | 26.00 | 37.10 |
| OccFusion (Ming et al., 2024) | C+R | 33.97 | 20.73 | 20.46 | 13.98 | 27.99 | 31.52 | 13.68 | 18.45 | 15.79 | 13.05 | 13.94 | 23.84 | 37.85 | 19.60 | 22.41 | 21.20 | 16.16 | 21.81 |
| OccFusion (Ming et al., 2024) | C+L | 44.35 | 26.87 | 26.67 | 18.38 | 32.97 | 35.81 | 19.39 | 22.17 | 24.48 | 17.77 | 21.46 | 29.67 | 39.01 | 21.94 | 24.90 | 26.76 | 28.53 | 40.03 |
| OccFusion (Ming et al., 2024) | C+L+R | 44.66 | 27.30 | 27.09 | 19.56 | 33.68 | 36.23 | 21.66 | 24.84 | 25.29 | 16.33 | 21.81 | 30.01 | 39.53 | 19.94 | 24.94 | 26.45 | 28.93 | 40.41 |
| GaussianFusionOcc** (Ours) | C | 37.05 | 22.43 | 22.46 | 14.19 | 28.66 | 29.93 | 15.10 | 17.08 | 18.22 | 9.71 | 18.21 | 25.12 | 37.19 | 20.59 | 23.39 | 23.65 | 22.91 | 32.55 |
| GaussianFusionOcc** (Ours) | C+R | 37.37 | 22.80 | 22.66 | 13.95 | 29.70 | 31.30 | 15.84 | 18.16 | 19.02 | 9.61 | 17.63 | 25.51 | 37.72 | 20.02 | 23.30 | 23.57 | 23.66 | 33.21 |
| GaussianFusionOcc*** (Ours) | L | 45.32 | 29.75 | 30.02 | 16.46 | 35.02 | 38.93 | 22.25 | 24.65 | 29.64 | 18.41 | 24.64 | 30.93 | 43.31 | 26.30 | 28.95 | 29.29 | 34.33 | 42.92 |
| GaussianFusionOcc (Ours) | C+L | 45.16 | 30.21 | 30.22 | 18.70 | 35.91 | 39.57 | 22.67 | 27.36 | 30.10 | 18.59 | 24.45 | 31.25 | 43.06 | 25.76 | 29.12 | 29.33 | 34.65 | 42.70 |
| GaussianFusionOcc (Ours) | C+L+R | 45.20 | 30.37 | 30.43 | 18.54 | 36.23 | 39.66 | 22.57 | 27.35 | 30.30 | 19.14 | 24.56 | 31.95 | 42.60 | 25.82 | 29.48 | 29.70 | 34.78 | 42.95 |

The superior performance of our proposed GaussianFusionOcc framework is clearly demonstrated in Table 1, where it outperforms a range of state-of-the-art methods across various categories of 3D semantic occupancy prediction on the nuScenes validation set (Caesar et al., 2020).

We compare our GaussianFusionOcc with several categories of existing methods:

**Comparison with planar-based models:** Our results indicate a significant improvement over planar-based methods such as BEVFormer (Li et al., 2024) and TPVFormer (Huang et al., 2023). While these models lift 2D image features to a Bird's-Eye View (BEV) or Tri-Perspective View (TPV), the object-centric nature of Gaussians in GaussianFusionOcc allows for a more efficient and flexible representation. GaussianFusionOcc outperforms planar-based methods across all semantic categories, with most pronounced gains for small or vertically complex classes like pedestrians or motorcycles.

**Comparison with grid-based models:** GaussianFusionOcc demonstrates improved performance over grid-based representations like OccFormer (Zhang et al., 2023). This is attributed to the sparse modeling of the scene with learnable Gaussians, which better allocates representational capacity to complex object shapes and details, particularly in capturing fine-grained structures and handling varying object scales.

**Comparison with existing gaussian-based models:** Even when compared to other Gaussian-based methods, such as GaussianFormer-2 (Huang et al., 2024a), GaussianFusionOcc demonstrates enhanced performance. This improvement can be attributed to the unique multi-sensor fusion mechanism employed in GaussianFusionOcc, which addresses the limitation of previous Gaussian-based approaches that rely solely on camera input. The superior results over GaussianFormer-2 highlight the effectiveness of our fusion framework in leveraging the benefits of the Gaussian representation for 3D semantic occupancy prediction with multi-sensor input.

**Comparison with sensor-fusion models:** Table 1 demonstrates that GaussianFusionOcc also surpasses sensor-fusion models such as M-CONet (Wang et al., 2023) and OccFusion (Ming et al., 2024). Notably, GaussianFusionOcc achieves this superior performance with a significantly lower number of parameters (79.96M) compared to M-CONet (137M) and OccFusion (114.97M). This suggests that our novel approach, utilizing a more effective representation and Gaussian-space sensor fusion, can surpass the performance of multi-sensor fusion techniques based on grid representations.

Overall, the results presented in Table 1 establish GaussianFusionOcc as a new state-of-the-art for 3D semantic occupancy prediction. Our model's superior performance across different architectural paradigms underscores the efficacy of its core design principles.

### 4.4 PERFORMANCE UNDER CHALLENGING SCENARIOS

Table 2: 3D semantic occupancy prediction results on rainy scenario subset of nuScenes validation set. All methods are trained with dense occupancy labels from (Wei et al., 2023). Modality notation: Camera (C), LiDAR (L), Radar (R).

| Method | Modality | IoU | mIoU | barrier | bicycle | bus | car | constr. veh. | motorcycle | pedestrian | traffic cone | trailer | truck | driveable surf. | other flat | sidewalk | terrain | man made | vegetation |
|---|---|---|---|---|---|---|---|---|---|---|---|---|---|---|---|---|---|---|---|
| OccFusion (Ming et al., 2024) | C | 31.10 | 18.99 | 18.55 | 14.29 | 22.28 | 30.02 | 10.19 | 15.20 | 10.03 | 9.71 | 13.28 | 20.98 | 37.18 | 23.47 | 27.74 | 17.46 | 10.36 | 23.13 |
| SurroundOcc (Wei et al., 2023) | C | 30.57 | 21.40 | 21.40 | 12.75 | 25.49 | 31.31 | 11.39 | 12.65 | 8.94 | 9.48 | 14.51 | 21.52 | 35.34 | 25.32 | 29.89 | 18.37 | 14.44 | 24.78 |
| OccFusion (Ming et al., 2024) | C+R | 33.75 | 20.78 | 20.14 | 16.33 | 26.37 | 32.39 | 11.56 | 17.08 | 11.14 | 10.54 | 13.61 | 22.42 | 37.50 | 22.79 | 29.50 | 17.58 | 17.06 | 26.49 |
| OccFusion (Ming et al., 2024) | C+L | 43.36 | 26.55 | 24.95 | 19.11 | 34.23 | 36.07 | 17.01 | 21.07 | 18.87 | 17.46 | 21.81 | 28.73 | 37.82 | 24.39 | 30.80 | 20.37 | 28.95 | 43.12 |
| OccFusion (Ming et al., 2024) | C+L+R | 43.50 | 26.72 | 25.30 | 18.71 | 33.58 | 36.28 | 17.76 | 22.44 | 20.80 | 15.89 | 22.63 | 28.75 | 39.28 | 22.72 | 30.78 | 20.15 | 28.99 | 43.37 |
| GaussianFusionOcc | C | 36.83 | 21.86 | 21.50 | 12.56 | 28.89 | 29.50 | 13.16 | 12.97 | 12.31 | 7.60 | 19.32 | 23.33 | 38.12 | 22.13 | 29.17 | 20.65 | 22.60 | 36.00 |
| GaussianFusionOcc | L | 43.80 | 28.40 | 27.16 | 16.79 | 36.99 | 38.09 | **21.08** | 16.09 | 25.48 | 17.60 | **26.67** | 30.37 | 40.36 | 24.30 | 33.59 | 22.47 | 33.28 | 44.16 |
| GaussianFusionOcc | C+R | 37.03 | 21.87 | 21.97 | 13.92 | 29.91 | 30.64 | 9.68 | 12.90 | 12.70 | 6.72 | 19.48 | 23.97 | 37.43 | 21.78 | 29.02 | 19.58 | 23.77 | 36.51 |
| GaussianFusionOcc | C+L | 44.28 | 29.19 | 28.10 | 19.84 | 36.28 | 38.90 | 18.11 | 21.13 | **26.14** | 17.95 | 25.79 | 29.92 | **41.72** | **27.35** | 34.99 | 22.85 | 33.84 | **44.10** |
| GaussianFusionOcc | C+L+R | **44.36** | **29.86** | **28.40** | **19.88** | **38.87** | **39.33** | 20.61 | **26.05** | 25.66 | **17.97** | 26.07 | **31.02** | 41.70 | 24.94 | **35.27** | **24.08** | **33.90** | 44.00 |

We further evaluate the performance of GaussianFusionOcc under challenging weather and lighting conditions on subsets of the nuScenes validation set, specifically focusing on rainy and nighttime scenarios. The results are presented in Table 2 and Table 3.

**Rainy scenario:** As shown in Table 2, GaussianFusionOcc (C+L+R) achieves an mIoU of 29.86% and IoU of 44.36%. Compared to single-modality variants, the fusion of camera, LiDAR, and radar consistently improves performance in rainy conditions. These results highlight the robustness of our multi-sensor fusion strategy to adverse weather, where cameras may struggle.

**Night scenario:** Table 3 shows performance in nighttime scenarios. LiDAR-only achieves the best results (43.00% IoU, 18.76% mIoU), demonstrating active sensing superiority in low-light conditions. Multi-modal configurations C+L and C+L+R show modest decreases of 0.10 and 0.31 mIoU respectively, likely due to noise-dominated gradients from sparse radar signals and cameras'

Table 3: 3D semantic occupancy prediction results on night scenario subset of nuScenes validation set. All methods are trained with dense occupancy labels from (Wei et al., 2023). Modality notation: Camera (C), LiDAR (L), Radar (R).

| Method | Modality | IoU | mIoU | barrier | bicycle | bus | car | constr. veh. | motorcycle | pedestrian | traffic cone | trailer | truck | driveable surf. | other flat | sidewalk | terrain | man made | vegetation |
|---|---|---|---|---|---|---|---|---|---|---|---|---|---|---|---|---|---|---|---|
| OccFusion (Ming et al., 2024) | C | 24.49 | 9.99 | 10.40 | 12.03 | 0.00 | 29.94 | 0.00 | 9.92 | 4.88 | 0.91 | 0.00 | 17.79 | 29.10 | 2.37 | 10.80 | 9.40 | 8.68 | 13.57 |
| SurroundOcc (Wei et al., 2023) | C | 24.38 | 10.80 | 10.55 | 14.60 | 0.00 | 31.05 | 0.00 | 8.26 | 5.37 | 0.58 | 0.00 | 18.75 | 30.72 | 2.74 | 12.39 | 11.53 | 10.52 | 15.77 |
| OccFusion (Ming et al., 2024) | C+R | 27.09 | 11.13 | 10.78 | 12.77 | 0.00 | 33.50 | 0.00 | 12.72 | 4.91 | 0.61 | 0.00 | 19.97 | 29.51 | 0.94 | 12.15 | 10.72 | 11.81 | 17.72 |
| OccFusion (Ming et al., 2024) | C+L | 41.38 | 15.26 | 12.74 | 13.52 | 0.00 | 35.85 | 0.00 | 15.33 | 13.19 | 0.83 | 0.00 | 23.78 | 32.49 | 0.92 | 14.24 | 20.54 | 23.57 | 37.10 |
| OccFusion (Ming et al., 2024) | C+L+R | 41.47 | 15.82 | 13.27 | **13.53** | 0.00 | 36.41 | 0.00 | 19.71 | 12.16 | **2.04** | 0.00 | 25.90 | 32.44 | 0.80 | 14.30 | 21.06 | 24.49 | 37.00 |
| GaussianFusionOcc | C | 32.05 | 11.28 | 6.13 | 4.41 | 0.00 | 30.40 | 0.00 | 12.51 | 3.06 | 0.43 | 0.00 | 21.55 | 28.91 | **3.54** | 13.10 | 13.71 | 16.12 | 26.58 |
| GaussianFusionOcc | L | **43.00** | **18.76** | **19.62** | 8.32 | 0.00 | 39.74 | 0.00 | 26.85 | 13.28 | 0.21 | 0.00 | **39.11** | 39.01 | 2.02 | **20.04** | **22.81** | 29.03 | **40.12** |
| GaussianFusionOcc | C+R | 33.08 | 12.31 | 9.18 | 7.68 | 0.00 | 32.05 | 0.00 | 14.14 | 5.75 | 0.00 | 0.00 | 20.89 | 31.17 | 2.52 | 12.87 | 15.10 | 17.29 | 28.35 |
| GaussianFusionOcc | C+L | 42.78 | 18.66 | 16.09 | 12.27 | 0.00 | **39.82** | 0.00 | 27.66 | 13.68 | 0.07 | 0.00 | 38.25 | **40.10** | 2.07 | 19.64 | 19.82 | **29.55** | 39.61 |
| GaussianFusionOcc | C+L+R | 42.51 | 18.45 | 12.15 | 11.47 | 0.00 | 39.77 | 0.00 | **29.58** | **15.27** | 0.04 | 0.00 | 37.08 | 37.13 | 2.58 | 19.94 | 20.63 | 29.42 | 40.10 |

limited signal in darkness. However, adding LiDAR or radar to camera-only significantly improves performance: camera-only achieves 32.05% IoU and 11.28% mIoU, while C+L gains +7.38 mIoU and C+R gains +1.03 mIoU. This demonstrates the robustness of multi-modal fusion when individual sensors are compromised, while leaving room for further research on utilizing more complex methods than simple concatenation in the fusion module.

The experimental results strongly suggest that GaussianFusionOcc shows significant performance gains in 3D semantic occupancy prediction, particularly in adverse weather and low-light scenarios.

## 4.5 EFFICIENCY ANALYSIS

Table 4: Efficiency comparison of multi-modal 3D semantic occupancy prediction on nuScenes validation set. Latency results marked with * were taken from the paper that introduced the model and were measured on a different GPU.

| Method | Modality | IoU | mIoU | Params | Memory (GB) | Latency (ms) |
|---|---|---|---|---|---|---|
| L-CONet (Wang et al., 2023) | L | 39.40 | 17.70 | - | 8.5 | - |
| M-CONet (Wang et al., 2023) | C+L | 39.20 | 24.70 | 137M | 24 | - |
| OccFusion (Ming et al., 2024) | C+R | 33.97 | 20.73 | 92.71M | 5.56 | 588* |
| OccFusion (Ming et al., 2024) | C+L | 44.35 | 26.87 | 92.71M | 5.56 | 591* |
| OccFusion (Ming et al., 2024) | C+L+R | 44.66 | 27.30 | 114.97M | 5.78 | 601* |
| GaussianFusionOcc | C | 37.05 | 22.43 | 71.39M | 2.44 | 282 |
| GaussianFusionOcc | C+R | 37.37 | 22.80 | 71.86M | 2.58 | 315 |
| GaussianFusionOcc | L | **45.32** | 29.75 | 34.14M | 0.49 | 179 |
| GaussianFusionOcc | C+L | 45.16 | 30.21 | 79.63M | 2.61 | 460 |
| GaussianFusionOcc | C+L+R | 45.20 | **30.37** | 79.96M | 2.90 | 480 |

Table 4 presents the efficiency comparison of multi-modal 3D semantic occupancy prediction methods, reporting the number of learnable parameters, memory usage, and latency. GaussianFusionOcc achieves state-of-the-art accuracy while substantially reducing computational and memory requirements compared to grid-based baselines. In the LiDAR-only setting, GaussianFusionOcc achieves 45.32% IoU and 29.75% mIoU, while requiring only 34.14 million parameters and 0.49GB of memory with inference time 179 ms. The GaussianFusionOcc further improves prediction performance in multi-modal settings, while staying significantly more efficient than comparable multi-modal methods. The memory and parameter savings of GaussianFusionOcc can be attributed to its adaptive, object-centric 3D Gaussian representation. This design enables the model to maintain a low memory footprint even as the number of sensor modalities increases. Latency measurements further highlight the practical advantages of GaussianFusionOcc, showing significantly faster inference than compared models. Detailed latency profiling reveals that feature extraction dominates the computational budget in both camera-only (81.8%) and full-sensor configurations (76.2%, distributed as 54.2% image, 21.6% LiDAR, 0.4% radar), while the four GaussianFusionBlocks contribute 13.7% and 15.6% respectively, and Gaussian-to-voxel splatting accounts for 4.5% and 8.1%. This distribution demonstrates that the primary bottleneck lies in the modality-specific encoders rather than in Gaussian refinement or splatting operations, validating the efficiency of our Gaussian-based representation. A complete breakdown of component-wise latency measurements

is provided in A.2. We report an additional ablation study on the number of Gaussians, the number of channels for feature representation, and the initialization strategy, in the appendix A.3.

### 4.6 QUALITATIVE ANALYSIS

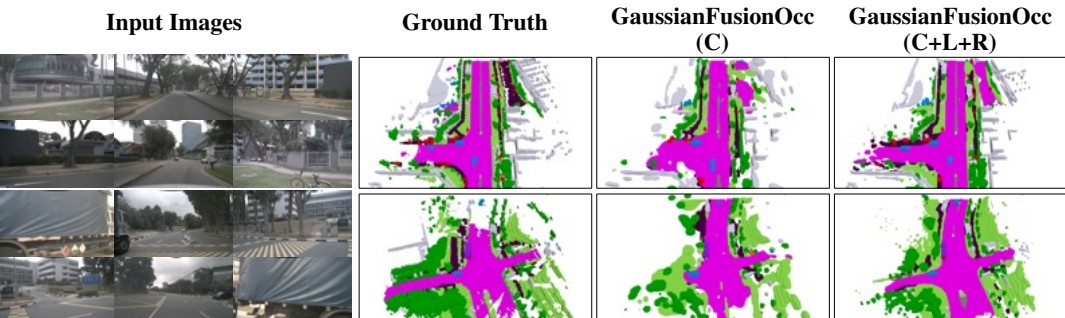

Figure 3: **Occupancy visualizations on nuScenes.** Our model is able to predict both comprehensive and realistic semantic 3D occupancy.

The analysis of the visualization results further underscores the benefits of multi-sensor fusion for 3D semantic occupancy prediction. In the Figure 3, the model incorporating camera, LiDAR, and radar data exhibits significantly better predictions in distant regions compared to the camera-only model, with LiDAR and radar effectively compensating for camera limitations. Figure 5 demonstrates the improvement in the allocation of the Gaussians when using the full sensor suite. Figure 4, comparing the performance of camera-only, camera+radar, and camera+LiDAR models in nighttime scenarios, clearly illustrates the critical role of additional sensors in overcoming the challenges posed by low-light conditions. The inclusion of both radar and LiDAR data leads to substantial improvements in the model's ability to perceive the environment under these conditions, as vision-centric approaches are known to perform poorly in nighttime scenarios due to the sensitivity of cameras to varying and limited illumination. LiDAR's 3D spatial awareness and radar's ability to detect dynamic objects contribute to more robust nighttime perception, with the most noticeable improvements in predicting distant dynamic objects and reducing close object hallucinations.

## 5 CONCLUSION

In this paper, we introduced GaussianFusionOcc, a novel and seamless sensor fusion framework for 3D semantic occupancy prediction that leverages the efficiency and flexibility of the 3D Gaussian representation. By fusing sensor information directly on learnable Gaussian primitives, it addresses critical limitations of grid-based methods, achieving state-of-the-art performance (30.37 mIoU and 45.20 IoU) while significantly reducing memory usage, number of parameters, and latency. The proposed modality-agnostic Gaussian encoder and fusion mechanism enable efficient integration of camera, LiDAR, and radar data, exploiting their complementary strengths. The experiments demonstrate the model's resilience in adverse weather and nighttime conditions, achieving significantly better results compared to previous state-of-the-art methods.

Even though sparse radar data provides complementary robustness in specific conditions, the concatenation-based fusion mechanism generally shows modest improvements with occasional degradation, suggesting that advanced strategies like cross-attention or confidence weighting could better balance heterogeneous sensors. Additionally, evaluating the model on other datasets could give valuable insight into the performance and robustness of the method. Finally, the framework's reliance on a predefined number of Gaussians limits adaptability in extremely sparse scenarios, potentially benefiting from dynamic pruning and densification mechanisms.

Despite these limitations, GaussianFusionOcc demonstrates that Gaussian-space fusion represents a promising paradigm shift for multi-modal perception, opening new research directions for primitive-based fusion in autonomous driving tasks beyond occupancy prediction, including object tracking, motion forecasting, and trajectory planning.

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

## A APPENDIX

### A.1 ADDITIONAL VISUALIZATIONS

The visualizations in Figure 4 and Figure 5 demonstrate the importance of sensor fusion for effective allocation of the semantic Gaussians, especially in low-light scenarios.

### A.2 INFERENCE LATENCY BREAKDOWN

To provide deeper insights into the computational characteristics of GaussianFusionOcc, we conducted comprehensive latency profiling of the inference pipeline across different sensor configurations. All measurements were performed on an NVIDIA RTX A6000 GPU with a batch size of 1, averaged over 100 inference runs on the nuScenes validation set. The pipeline is decomposed into three main stages: (1) modality-specific feature extraction using the image backbone (ResNet101-DCN + FPN), LiDAR encoder (VoxelNet + FPN), and radar encoder (PointPillars), (2) the four sequential GaussianFusionBlocks that iteratively refine Gaussian properties through modality-agnostic deformable attention and fusion, and (3) the Gaussian-to-voxel splatting module that transforms the refined Gaussian representation into the final dense occupancy prediction.

In the camera-only variant, feature extraction from the six surround-view cameras constitutes the dominant computational cost at approximately 81.8% of total inference time. The four GaussianFusionBlocks collectively account for roughly 13.7%, while the Gaussian-to-voxel splatting module contributes 4.5%. This distribution reveals that the computational bottleneck lies primarily in the image feature extraction backbone rather than in the Gaussian refinement or splatting operations. The relatively modest contribution of the GaussianFusionBlocks highlights the efficiency of our modality-agnostic deformable attention mechanism. Despite processing 6,400 Gaussians through four refinement iterations with 128-channel feature representations, the Gaussian refinement stage adds minimal overhead. This efficiency stems from the sparse nature of Gaussian-based representations, where attention operations focus computational resources on occupied regions rather than processing dense grids uniformly. The Gaussian-to-voxel splatting module's small contribution further validates the computational advantages of our approach compared to dense voxel-based methods that require expensive 3D convolutions throughout the entire pipeline.

When integrating camera, LiDAR, and radar inputs, the latency distribution shifts to reflect the additional computational cost of multi-modal feature extraction and fusion. Collectively, feature extraction from all modalities accounts for approximately 76.2% of total inference time, distributed as follows: image extraction dominates at 54.2%, LiDAR extraction contributes 21.6%, while radar processing introduces negligible overhead at 0.4%. This demonstrates that radar is computationally attractive for enhancing robustness without significant latency penalties. The GaussianFusionBlocks' contribution increases from 13.7% to 15.6% in the multi-modal setting, reflecting the additional computational cost of extracting per-Gaussian features from multiple sensor modalities through deformable attention and fusing them via the concatenation-based fusion module. This increase is modest considering that the blocks now process features from three distinct sensor types with different spatial resolutions and representational characteristics. The Gaussian-to-voxel splatting increases from 4.5% to approximately 8.1%, suggesting that splatting from richer multi-modal Gaussian representations with more refined geometric and semantic properties requires additional computation, yet remains significantly more efficient than grid-based alternatives.

| Ground Truth | GaussianFusionOcc (C) | GaussianFusionOcc (C+R) | GaussianFusionOcc (C+L) |
|---|---|---|---|

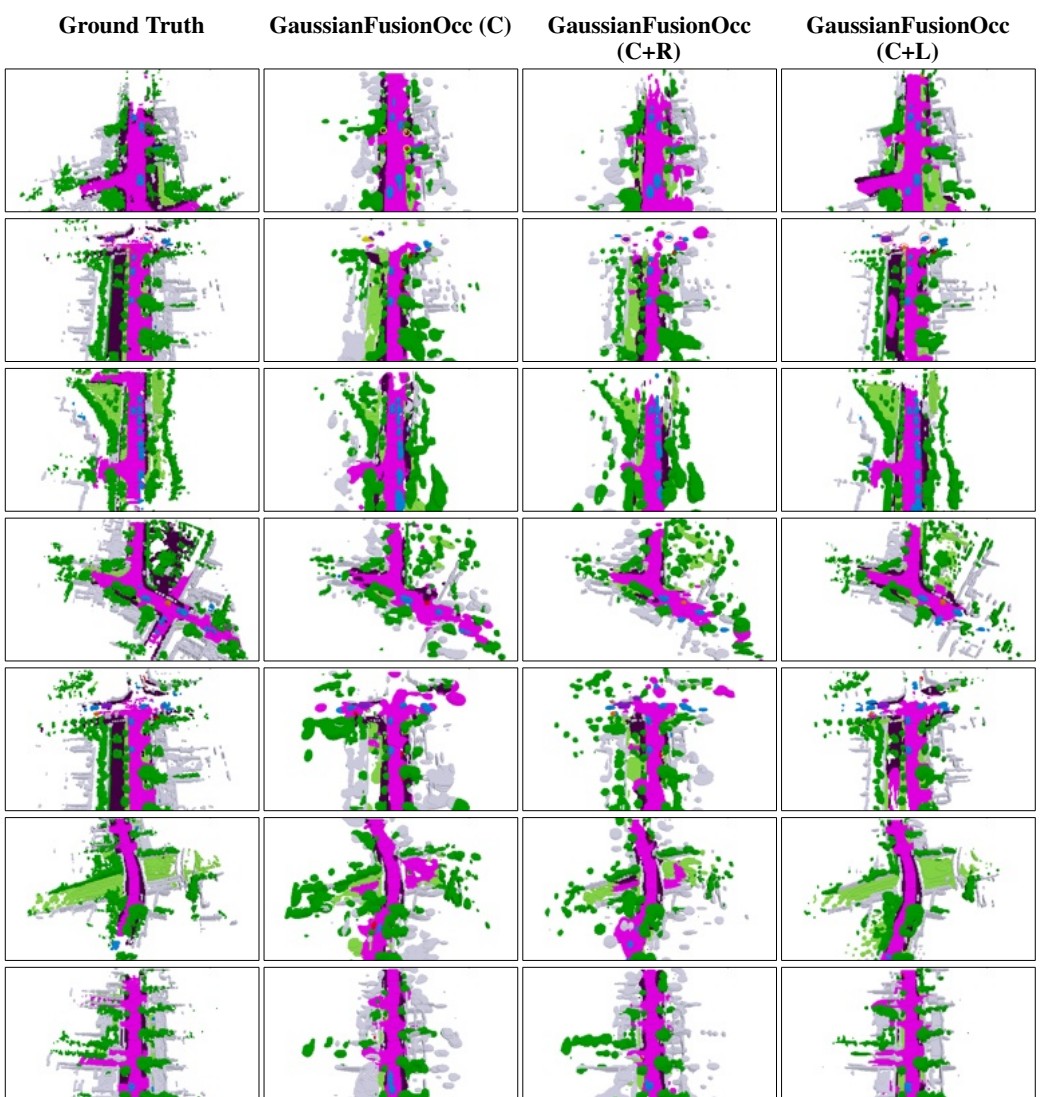

Figure 4: **Occupancy visualizations on nuScenes night scenes.** Visualization shows the importance of additional sensors in low-light conditions.

The profiling results reveal several important insights. First, feature extraction from modality-specific encoders constitutes the primary computational bottleneck across all configurations (81.8% camera-only, 76.2% full-suite), indicating that further optimization efforts should focus on efficient backbone architectures rather than the Gaussian-based components. Second, the GaussianFusion-Blocks demonstrate remarkable efficiency, adding only 13.7-15.6% overhead despite performing complex multi-modal fusion and iterative refinement operations. This validates our design choice of sparse Gaussian representation combined with deformable attention, which avoids the computational redundancy inherent in dense grid-based approaches. Third, the Gaussian-to-voxel splatting module's minimal contribution (4.5-8.1%) confirms that our splatting-based final prediction stage is substantially more efficient than methods that maintain dense volumetric representations throughout the entire pipeline.

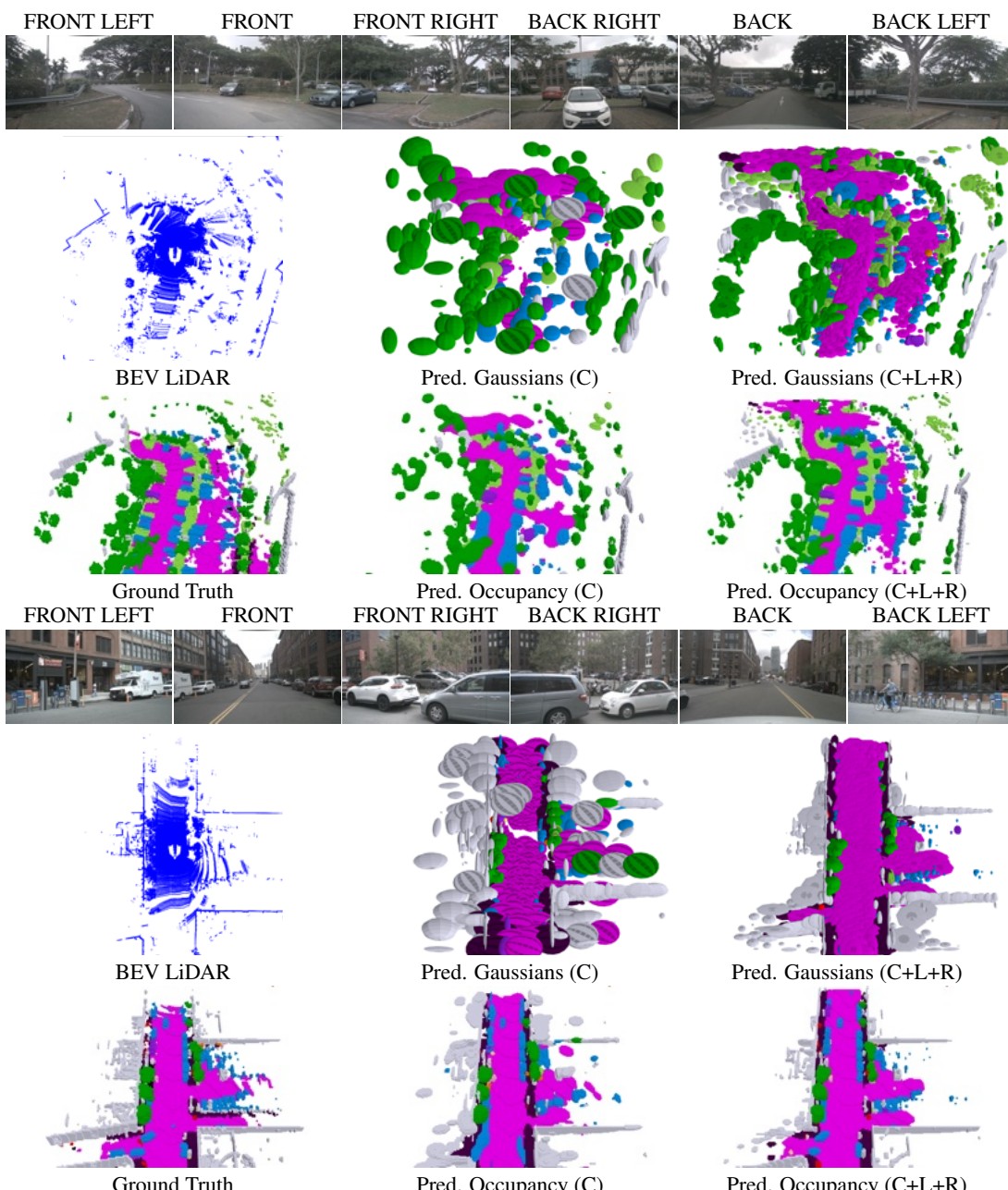

Figure 5: **Occupancy and Gaussian representation visualizations on nuScenes dataset.** Visualizations demonstrate the importance of additional sensors for the allocation of the Gaussians and final occupancy prediction.

## A.3 ABLATION STUDY

We conduct ablation studies to analyze the impact of key design choices and components of our GaussianFusionOcc framework. The latency and memory are tested on an NVIDIA RTX A6000 GPU for all the experiments.

**Number of Gaussians:** Table 5 shows the influence of the number of Gaussians on efficiency and performance. We observe improvement in performance as the number of Gaussians increases. This is due to the enhanced ability to represent finer details with more Gaussians. The number of

Table 5: Ablation on number of Gaussians. The increased number of Gaussians improves the performance, without significantly increasing memory usage and the number of parameters, but it increases the latency.

| Method | Modality | IoU | mIoU | Gaussians | Params | Memory (GB) | Latency (ms) |
|---|---|---|---|---|---|---|---|
| GaussianFusionOcc | C+L | 45.16 | 30.21 | 6400 | 79.63M | 2.61 | 460 |
| GaussianFusionOcc | C+L | 45.74 | 30.83 | 25600 | 80M | 2.62 | 547 |

Table 6: Ablation on number of channels for feature representation. The increased number of channels improves the performance, but significantly increases the number of parameters and memory usage.

| Method | Modality | IoU | mIoU | Channels | Params | Memory (GB) | Latency (ms) |
|---|---|---|---|---|---|---|---|
| GaussianFusionOcc | C+L+R | 45.20 | 30.37 | 128 | 79.96M | 2.90 | 480 |
| GaussianFusionOcc | C+L+R | 45.69 | 30.85 | 192 | 115M | 5.41 | 486 |

Table 7: Ablation on initialization strategy. Probabilistic initialization is taken from Huang et al. (2024a).

| Method | Modality | IoU | mIoU | Initinalization | Params | Memory (GB) | Latency (ms) |
|---|---|---|---|---|---|---|---|
| GaussianFusionOcc | C+L+R | 45.31 | 30.07 | Random | 79.78M | 2.62 | 468 |
| GaussianFusionOcc | C+L+R | 45.20 | 30.37 | Learnable | 79.96M | 2.90 | 480 |
| GaussianFusionOcc | C+L+R | 44.52 | 30.13 | Probabilistic | 79.85M | 3.05 | 844 |

parameters and memory usage are not significantly increased because they are mostly influenced by sensor-specific encoders, which are not influenced by the number of Gaussians.

**Number of channels:** The influence of the number of channels used for extracted sensor features and per-Gaussian features is demonstrated in Table 6. An Increased number of channels improves the prediction performance with a significant efficiency degradation. The increase in memory and parameter number can be attributed to the increased size of sensor feature extractors, as they are also influenced by the number of channels.

**Initialization strategy:** We report the influence of initialization strategy on the performance and efficiency in Table 7. Learnable initialization shows the highest mIoU with a slightly higher number of parameters, memory usage, and latency, compared to random initialization. Probabilistic initialization, proposed by GaussianFormer-2 (Huang et al., 2024a), degrades the performance of the model while significantly slowing down the inference.

### A.4 EVALUATION METRICS

To evaluate our method and compare the results with other state-of-the-art methods, we use Intersection over Union (IoU) and mean Intersection over Union (mIoU) metrics:

$$IoU = \frac{TP_{\neg c_0}}{TP_{\neg c_0} + FP_{\neg c_0} + FN_{\neg c_0}} \tag{7}$$

$$mIoU = \frac{1}{|C|} \sum_{c \in C} \frac{TP_c}{TP_c + FP_c + FN_c} \tag{8}$$

where TP, FP, FN denote the number of true positive, false positive, and false negative predictions, and C, $c_0$ denote the set of classes without the empty class, and the empty class, respectively.

### A.5 THE USE OF LARGE LANGUAGE MODELS

In the preparation of this paper, we utilized Perplexity as a general-purpose writing assist tool. Specifically, the LLM was employed to improve grammar, refine wording, and enhance the overall clarity and flow of the written content. The LLM was not involved in research ideation, methodology design, experimental planning, or the interpretation of results. All scientific contributions, including

the conceptualization of GaussianFusionOcc, the development of the fusion framework, experimental design, and analysis of results, were entirely conducted by the authors. The authors take full responsibility for all content presented in this work, including any text that was refined with LLM assistance.

