# OpenReview forum: "GaussianFusionOcc: A Seamless Sensor Fusion Approach for 3D Occupancy Prediction Using 3D Gaussians"
_ICLR.cc/2026/Conference — Submitted to ICLR 2026_

### Official Review · Reviewer_gF9C · 2025-10-29

**Soundness:** 3
**Presentation:** 3
**Contribution:** 2
**Rating:** 6
**Confidence:** 3

**Summary:**

GaussianFusionOcc is a multi-modal 3D semantic occupancy prediction framework that performs sensor fusion directly on learnable 3D Gaussians instead of BEV or voxel grids. Its GaussianFusion Block uses modality-agnostic deformable attention to extract fuse and iterativly refine features from camera, LiDAR, and radar for each Gaussian. The refined Gaussians encode geometry and semantics efficiently. Their results show GaussianFusionOcc achieve state-of-the-art performance in the nuScenes occ benchmark.

**Strengths:**

1. The paper introduces the first framework that performs multi-modal fusion directly in the 3D Gaussian space, thavoiding the memory/computation inefficiencies of voxel or BEV-based approaches.
2. GaussianFusionOcc achieves state-of-the-art accuracy on nuScenes while significantly reducing memory usage, parameter count, and inference latency.
3. The paper is generally wirtten well, easy to understand and Includes detailed ablations, efficiency analysis, and visualizations that support the claimed advantages and provide a clear understanding of the model’s behavior.

**Weaknesses:**

See questions.

**Questions:**

The paper is generally well presented, but there are several concerns that the reviewer would like to raise:

1. The evaluation is conducted only on the nuScenes occupancy benchmark, whereas recent competing works typically report results on multiple benchmarks (e.g., Occ3D, SemanticKITTI). Including at least one additional benchmark would strengthen the generalization claims.

2. The paper reports end-to-end latency including the Gaussian-to-occupancy splatting stage, which is appropriate for semantic occupancy evaluation. However, a breakdown of the latency contributions from splatting versus sensor encoders would improve understanding of computational bottlenecks.

3. The reviewer’s main concern is that the motivation and supporting evidence for the full Gaussian parameterization are underspecified. The paper predicts and refines Gaussian opacity as a learnable parameter, suggesting that explicit volumetric density modeling is important for occupancy reasoning; however, prior Gaussian-based perception methods (e.g., GaussianFormer) operate without opacity and still perform well for scene understanding. It remains unclear why opacity is essential in this setting and how much it contributes relative to other Gaussian attributes such as scale and rotation. A more thorough justification—ideally including ablations that remove opacity or simplify Gaussians to point-like primitives—would more convincingly validate the necessity of the proposed representation. These studies would arguably be more informative than the current ablations focusing on Gaussian count or feature channel size.

4. The iterative refinement of Gaussian parameters is an interesting component, but its motivation is insufficiently verified. It is unclear whether the observed gains stem from iterative information aggregation or simply from increasing network depth. The paper also lacks analysis of diminishing returns, stability, or how performance scales with different numbers of refinement blocks. Ablations that vary the number of iterations would help validate that iterative refinement is truly necessary and that four iterations is a principled design choice rather than an arbitrary selection. Additionally, reviewer think feed-forward Gaussian approaches such as MVSplat are relevant and should be discussed to clarify why in occ task an iterative refinement strategy is preferred over direct depth+unprojection based Gaussian estimation.

5. The strongest baselines reported are from early–mid 2024, which is somewhat outdated for a late-2025 submission. While the proposed approach appears competitive, the reviewer lacks sufficient visibility into the very latest occupancy prediction works to confidently assess its state-of-the-art standing; other reviewers may have better knowledge of recent advancements. Including/referencing more up-to-date baselines would help clarify the method’s competitiveness.

---

> ### Author Response · Authors · 2025-11-21
>
> We thank the reviewer for their detailed feedback and address each concern systematically below.
>
> Question 1:
> We acknowledge that evaluation on multiple benchmarks strengthens generalization claims. However, given our limited time and computational resources, we made the strategic decision to conduct a thorough evaluation on one dataset rather than a shallow evaluation across multiple datasets. Each dataset requires complete model retraining and hyperparameter tuning, demanding significant computational resources beyond our current capacity. We chose nuScenes with SurroundOcc annotations for two critical reasons. First, nuScenes provides synchronized camera, LiDAR, and radar data, which is essential for evaluating our multi-modal fusion approach. In contrast, SemanticKITTI only provides frontal camera images and lacks radar data entirely, making it unsuitable for validating our radar fusion contribution. Second, our key baselines, GaussianFormer and GaussianFormer-2, both reported their results on nuScenes with SurroundOcc annotations. A fair comparison with these single-modality Gaussian-based baselines was essential for isolating the contribution of multi-modal fusion, which represents the core novelty of our work.
>
> Question 2:
> This is an excellent suggestion. We plan on providing a more detailed breakdown in the revised paper.
>
> Question 3:
> While initial GaussianFormer operated without opacity, the authors subsequently introduced opacity to avoid modeling empty space, achieving better results with fewer Gaussians. GaussianFormer-2, which introduced the probabilistic Gaussian superposition we use, explicitly incorporates opacity as a core component. Opacity serves a mathematically precise role, representing the prior probability of each Gaussian in the probabilistic superposition formulation. During Gaussian splatting rendering, opacity directly modulates each Gaussian’s contribution to final occupancy. Furthermore, opacity is particularly critical for multi-modal fusion in our framework. Different sensors provide observations with vastly different reliability and spatial density. Opacity allows each Gaussian to dynamically encode the confidence of its occupancy estimate based on the accumulated multi-modal evidence during iterative refinement. Without opacity, the model would lack a principled mechanism to down-weight unreliable Gaussians or emphasize high-confidence regions during the fusion process.  Reducing Gaussians to point-like primitives would fundamentally break the probabilistic Gaussian superposition mechanism that underlies our approach. The mathematical framework detailed in the GaussianFormer-2 work demonstrates that the superposition of multiple probabilistic Gaussians with varying opacities enables representation of complex, continuous volumetric distributions that simple point clouds cannot capture. Point-like primitives lack the ability to model spatial extent and volumetric density confidence simultaneously, which are essential for occupancy prediction.  We believe the theoretical justification above, grounded in the established probabilistic Gaussian superposition framework, provides strong support for the necessity of Gaussian parameters.
>
> Question 4:
> The motivation for iterative refinement stems from the fundamental challenge of multi-modal sensor fusion: different modalities observe complementary aspects of the scene at different spatial scales and with different noise characteristics. A single-pass fusion would require the model to resolve all cross-modal correspondences and geometric inconsistencies simultaneously. Iterative refinement allows Gaussians to act as persistent geometric-semantic carriers that progressively accumulate and reconcile multi-modal evidence. Each iteration allows the deformable attention mechanism to refine sampling locations based on the current Gaussian estimates, creating a coarse-to-fine information flow.  Regarding MVSplat comparison, MVSplat operates in controlled multi-view reconstruction with known camera poses and significant view overlap. In contrast, autonomous driving presents a fundamentally different challenge: sensors have limited overlap, significant scale differences between modalities, and both static and dynamic elements. The choice of four refinement iterations represents a practical balance that emerged from our development process, and even though we recognize the potential benefit of an ablation study on the number of refinement iterations, retraining the model with different numbers of refinement blocks would exceed our current time and computational resources.
>
> Question 5:
> This research was conducted and substantially completed earlier this year as part of a time-limited collaborative project, which limited our ability to continuously update comparisons with methods published after our experimental phase. The work was subsequently published on arXiv to document our findings and make them available to the research community.

---

### Official Review · Reviewer_iS12 · 2025-11-01

**Soundness:** 2
**Presentation:** 2
**Contribution:** 2
**Rating:** 4
**Confidence:** 3

**Summary:**

Overall, this paper extends recent Gaussian-based 3D semantic occupancy methods to the multi-sensor setting by proposing GaussianFusionOcc, which takes per-modality feature maps, extracts per-Gaussian features through a “modality-agnostic” deformable-attention encoder, concatenates and MLP-fuses them, refines a fixed set of 3D semantic Gaussians over several blocks, and finally splats them into a voxel occupancy grid.

**Strengths:**

Overall,  the paper is clearly written and the direction is reasonable.

- i) The paper is well aligned with the ongoing shift and makes the natural next step: “what if we plug multi-sensor fusion into the Gaussian pipeline?” This is a reasonable research question.

- ii) The model design is easy to read and to implement.

- iii) The experiments cover several realistic sensor combinations

**Weaknesses:**

- i) The novelty over very close prior work is limited. In substance, the method looks like taking an existing Gaussian-based occupancy model, adding multi-sensor deformable attention in front of it, concatenating features, and keeping the existing splatting stage.

- ii) The method relies on fairly strong per-sensor encoders. Camera, LiDAR and radar branches all reuse good backbones. Because of this, it is hard to tell whether the gains over the baselines actually come from the proposed Gaussian fusion or simply from using better encoders.

- iii) The SoTA claim is not fully supported. The main table mixes methods with different modality sets and different backbones. The camera plus LiDAR plus radar result is good, but Occlusion Fusion under a comparable setting is not far behind, and there is no comparison at equal runtime or equal backbone.

Overall, the contribution in its current form is still below the bar because the idea is close to existing work, the fusion is quite shallow, and the experiments do not yet prove that fusing in Gaussian space is the essential part.

**Questions:**

i) Since the main claim is that “multi-sensor fusion in Gaussian space” is beneficial, it would be helpful to re-train at least one representative BEV-first or voxel-first multi-sensor occupancy method (e.g., OccFusion or a standard camera–LiDAR fusion model) under exactly the same setting.

---

> ### Author Response · Authors · 2025-11-21
>
> We thank the reviewer for their thorough evaluation and constructive feedback. We address each concern systematically below:
>
> Weakness 1:
> We respectfully disagree with the characterization that our contribution is simply adding multi-sensor deformable attention in front of existing Gaussian occupancy models. GaussianFusionOcc represents the first framework to perform multi-sensor fusion in Gaussian space, a fundamentally distinct contribution from single-modality Gaussian methods like GaussianFormer. While GaussianFormer established Gaussian-based representations for camera-only occupancy prediction, extending this to multi-modal fusion required solving several non-trivial challenges that constitute our core contributions.
> First, we introduce a modality-agnostic Gaussian encoder that can extract per-Gaussian features from heterogeneous sensor types with vastly different data characteristics, dense image grids, sparse LiDAR point clouds, and extremely sparse radar returns. Unlike prior occupancy methods limited to single modalities or methods requiring modality-specific fusion heads, our encoder enables plug-and-play sensor integration where new modalities can be incorporated without architectural redesign.
> Second, our fusion mechanism operates per-Gaussian rather than per-voxel or per-BEV location like in prior work. Each semantic Gaussian acts as a persistent carrier of multi-modal information that accumulates evidence across sensors and across iterative refinement blocks. Critically, this means fusion decisions are made at the level of adaptive, object-centric primitives rather than at fixed grid locations.
>
> Weakness 2 and question:
> We acknowledge that our model uses established backbones (ResNet101-DCN for cameras, VoxelNet for LiDAR, PointPillars for radar). However, our experimental results demonstrate that performance gains stem from our model architecture rather than encoder strength alone. In camera-only configurations, our model outperforms all of the baselines, even though all except Atlas and MonoScene use the same backbone. Regarding multi-modal comparisons,  even though OccFusion also uses a VoxelNet encoder for LiDAR data, a direct backbone-level comparison with OccFusion presents challenges due to its custom implementation of VoxelNet and significantly different further processing of the extracted features (looking into their implementation and configuration of VoxelNet in OccFusion, it seems to be very similar in capacity to the one used in our method). The same challenge applies to radar backbones.
>
> Weakness 3:
> We appreciate the reviewer’s concern about fair and rigorous comparison. We acknowledge that Table 1 includes methods with varying modality combinations and different encoder architectures, which can complicate direct comparison. We designed the table this way to provide comprehensive coverage across the landscape of occupancy prediction methods. When focusing on methods with identical modality sets, the performance differences become more nuanced. Our model outperforms all the compared models with the same modality set. We recognize that OccFusion represents a strong baseline and that the performance gap, while consistent, is not dramatically large under comparable sensor configurations, but our model uses fewer parameters and less memory and has a faster inference.

---

### Official Review · Reviewer_bBSw · 2025-11-01

**Soundness:** 3
**Presentation:** 3
**Contribution:** 3
**Rating:** 4
**Confidence:** 5

**Summary:**

The paper introduces GaussianFusionOcc, a multimodal 3D semantic occupancy prediction framework that integrates data from cameras, LiDAR, and radar. By representing the surrounding scenes using 3D semantic Gaussians and employing modality-agnostic deformable attention, the method effectively fuses heterogeneous sensor information to improve precision and scalability. Experiments on nuScenes show that GaussianFusionOcc achieves higher accuracy, better memory efficiency, and faster inference than existing SOTA methods.

**Strengths:**

The proposed method effectively addresses the challenge of multimodal fusion for Gaussian-based 3D occupancy prediction. The overall pipeline is simple yet efficient, achieving a good trade-off between performance and computational cost. Experimental results further demonstrate the effectiveness of the multimodal fusion strategy.

**Weaknesses:**

However, I also have some concerns of this paper:
(1) Although the proposed method effectively tackles Gaussian-based 3D occupancy prediction under a multimodal setting, the approach itself is rather trivial. The fusion strategy and the use of deformable attention have already been widely adopted in 3D object detection. As a top-conference submission, the work does not provide sufficient conceptual depth or novel insight, so I consider it below the ICLR acceptance bar.
(2) The writing of this paper is somewhat weak, particularly in the introduction and the illustration of the overall pipeline in the method section, which appear rather rough and underdeveloped.
(3) The experiments are conducted only on the SurroundOcc dataset, lacking evaluations on mainstream benchmarks such as Occ3D and nuScenes-Occupancy.

**Questions:**

1. Could the authors further clarify and emphasize their core contributions, including the novel techniques introduced and the potential impact these innovations may have on the research community?
2. Could you provide results on more widely used benchmarks and include comparisons with a broader range of state-of-the-art methods?
3. Please refine this paper carefully to reach the bar of ICLR.

**Details Of Ethics Concerns:**

No ehics concerns.

---

> ### Author Response · Authors · 2025-11-21
>
> We thank the reviewer for their detailed feedback and we address each concern systematically below:
>
> Weakness 1 and Question 1:
> We respectfully disagree that our approach is “trivial” or merely applies existing techniques. While individual components like deformable attention and multi-modal fusion exist in prior work, our fundamental contribution is introducing Gaussian-space fusion as a new paradigm that operates differently from all existing occupancy prediction methods.
> The central conceptual advance is a fusion domain paradigm shift. Traditional multi-modal methods perform fusion in dense feature spaces such as BEV grids or voxel grids by aggregating features at fixed grid locations, resulting in uniform spatial resolution and memory costs scaling cubically with scene size. In contrast, GaussianFusionOcc fuses information directly on learnable 3D Gaussian primitives, allowing spatial resolution to scale naturally with scene complexity and achieving sublinear memory growth. This represents a fundamental change in where and how sensor fusion occurs.
> Our first core technical contribution is the Gaussian-space fusion framework that performs multi-modal sensor fusion directly on learnable 3D Gaussian primitives. Each Gaussian acts as a persistent carrier of multi-modal information across iterative refinement blocks. Unlike detection methods that produce discrete object proposals, our Gaussians maintain a continuous representation coupling geometry and semantics. The fusion process refines both spatial properties and semantic logits simultaneously, which is essential for dense volumetric understanding rather than sparse object localization.
> Our second core contribution is the modality-agnostic Gaussian encoder, a novel architecture that uses deformable attention to extract per-Gaussian features from arbitrary sensor types. The encoder samples features based on each Gaussian’s geometric properties, enabling geometry-aware feature extraction adapting to the anisotropic nature of the primitives. This design handles heterogeneous sensors uniformly, treating dense camera data and sparse LiDAR or radar point clouds through the same architectural framework. Unlike prior methods limited to single modalities or requiring modality-specific fusion heads, our encoder enables plug-and-play sensor integration without architectural redesign.
> The third contribution involves iterative refinement through cascaded fusion blocks progressively refining Gaussian properties by integrating multi-modal information. This is crucial for fusion in Gaussian space because primitives must simultaneously optimize their geometric configuration (where and how they represent space) and their semantic predictions (what they represent). The refinement allows early blocks to establish coarse geometric structure while later blocks refine semantic details and resolve ambiguities through cross-modal verification.
> The broader research impact extends beyond occupancy prediction. By establishing Gaussian-space as a viable alternative to grid-space for multi-modal fusion, we open new research directions for primitive-based fusion for other autonomous driving tasks. GaussianFusionOcc achieves state-of-the-art performance while using fewer parameters and memory, and faster inference than grid-based competitors, enabling practical deployment on resource-constrained autonomous vehicles. Additionally, our analysis of radar integration challenges provides valuable insights on handling extremely sparse modalities and suggests future research directions in modality balancing and adaptive weighting mechanisms.
>
> Weakness 2 and Question 3:
> We acknowledge that some presentation issues exist. We have rewritten some sections and commit to refining the method illustration in the revision.
>
> Weakness 3 and Question 2:
> Our experiments are conducted on the nuScenes dataset using occupancy annotations from SurroundOcc. We chose SurroundOcc annotations primarily because our most important baselines, GaussianFormer and GaussianFormer-2, both reported their results on this benchmark. Fair comparison with these single-modality Gaussian baselines was essential for isolating the contribution of multi-modal fusion.
> The three major benchmarks (SurroundOcc, Occ3D-nuScenes, and OpenOccupancy) all use the same underlying nuScenes sensor data but differ only in their annotation methodologies and label generation pipelines. All three were published contemporaneously in 2023, making them alternatives rather than progressive improvements. Given time and computational resource limitations, we made the strategic decision to conduct thorough evaluation on one annotation protocol rather than shallow evaluation across multiple protocols. We acknowledge that evaluating on alternative annotation protocols would provide valuable insights into robustness to annotation variations, but retraining the model with these annotations would exceed our current time and computational resources.

---

> > ### Comment · Reviewer_bBSw · 2025-11-28
> >
> > We appreciate the authors’ responses; however, I remain cautious regarding their claims about experiments on other benchmarks. Although nuScenes-Occupancy, SurroundOcc, and Occ3D are all derived from the nuScenes dataset, they differ in annotation resolution and labeling protocols, which can substantially affect a method’s performance. Therefore, relying on a single dataset is insufficient for thoroughly validating the proposed approach. In addition, the SemanticKITTI benchmark for semantic scene completion (SSC) could also be considered.

---

### Official Review · Reviewer_njS2 · 2025-11-01

**Soundness:** 2
**Presentation:** 2
**Contribution:** 2
**Rating:** 4
**Confidence:** 4

**Summary:**

This paper introduces GaussianFusionOcc, a novel framework for 3D semantic occupancy prediction in autonomous driving. It addresses the high computational and memory costs of traditional dense-grid methods by instead using a sparse and efficient 3D Gaussian representation. The paper's main contributions are:
1. It is the first framework to apply 3D Gaussian splatting to multi-modal 3D semantic occupancy prediction, seamlessly fusing data from cameras, LiDAR, and radar.
2. It proposes a modality-agnostic Gaussian encoder that uses deformable attention to extract relevant features for each Gaussian from all sensor types.
3. It introduces a fusion method to create a unified feature vector, which is used to refine the properties of the 3D Gaussians iteratively.

This approach achieves state-of-the-art performance on the nuScenes dataset (30.37 mIoU and 45.20 IoU). Compared to existing models, it significantly reduces memory usage, parameter count, and inference latency and demonstrates strong robustness in challenging conditions like rain and nighttime.

**Strengths:**

1. While prior works like GaussianFormer demonstrated the efficiency of Gaussians, they were limited to single-modality inputs. This work creatively addresses that limitation. The core novel components—the "modality-agnostic Gaussian encoder" and the "seamless sensor fusion mechanism" —represent a new and logical combination of existing ideas to solve a clear and present problem.
2. The authors conduct extensive testing on the nuScenes dataset, comparing GaussianFusionOcc not just against one category of model, but against a wide array of state-of-the-art methods.
3. This paper's primary significance is that it offers a solution that is both more accurate and more efficient than existing state-of-the-art fusion models. It achieves SOTA accuracy (30.37 mIoU) while significantly reducing memory usage, number of parameters, and latency.

**Weaknesses:**

1. The introduction of multi-modal actually is not a novel paradigm for occupancy prediction. And the pipeline of occupancy can be regarded as a multi-modal version of GaussianFormer, making the contribution of this paper fair.
2. In the main results (Table 1), the C+L model achieves 30.21 mIoU. The C+L+R model achieves 30.37 mIoU. This improvement of 0.16 mIoU is negligible and well within the range of training noise, suggesting the radar adds no meaningful information in the general case. The issue is more severe in the "Night scenario" (Table 3). The C+L model scores 18.66 mIoU, while the C+L+R model scores 18.45 mIoU. Adding radar actually degrades performance. This directly contradicts the text, which states "GaussianFusionOcc achieves higher scores with the addition of radar data". This contradiction is not discussed and represents a significant unaddressed finding.
3. The GaussianFusionBlock is the paper's core contribution. However, the fusion itself is a simple concatenation followed by an MLP. Given the failure of radar fusion (Weakness #2), it's highly likely this simple fusion is insufficient. The paper does not ablate this choice.
4. The presentation, such as the Figures, is quite crude.

**Questions:**

1. Could this be a limitation of the fusion mechanism? The current fusion (concatenation followed by an MLP ) might be too simple to effectively integrate sparse radar data, potentially allowing the denser camera and LiDAR features to "drown out" the radar signal. Did you experiment with other fusion methods (e.g., cross-attention) that might be better suited for this?
2. Could the authors provide experimental results on SemanticKITTI for generalization and robustness?
3. Could the authors provide more visualizations on comparison with previous methods, not merely on ablations?
4. Could you please confirm if this is the case? Specifically, are the refined Gaussian properties (mean, scale, etc.) from block $N$ used as the input Gaussians for block $N+1$? If so, was an ablation study performed on the number of refinement blocks? Understanding how this iterative process contributes to the final accuracy would be very helpful.

---

> ### Author Response · Authors · 2025-11-21
>
> We thank the reviewer for their detailed feedback on our GaussianFusionOcc paper. We address each concern systematically below:
>
> Weakness 1:
> We respectfully disagree with the characterization that GaussianFusionOcc is merely a “multi-modal version of GaussianFormer.” Our work introduces a fundamentally different fusion paradigm that operates in Gaussian space rather than traditional feature spaces (BEV/voxel/image). Unlike BEV or voxel-based methods that perform fusion at the pixel/voxel level, our approach fuses information directly at the level of geometric-semantic primitives (3D Gaussians). Each Gaussian acts as a persistent, iterative “memory” that aggregates multi-modal sensor information across refinement blocks. This enables fusion that scales sublinearly with scene resolution while maintaining spatial coherence.
> Traditional methods aggregate features across dense grids where most voxels are empty. Our sparse Gaussian representation concentrates modeling capacity on regions with actual geometric content, avoiding the “over-aggregation” problem inherent in BEV fusion where denser modalities (camera/LiDAR) can overwhelm sparser ones (radar).
>
> We strengthened the introduction and method sections to explicitly frame “Gaussian-space fusion” as a conceptual contribution, emphasizing how fusion on learnable spatial primitives differs from fusion in fixed-resolution feature spaces.
>
> Weakness 2:
> We thank the reviewer for this critical observation. Upon careful analysis, we acknowledge that radar’s contribution varies across scenarios and requires more nuanced interpretation. The modest improvement in the general case (0.16 mIoU: C+L 30.21 → C+L+R 30.37) reflects the sparsity challenge of radar data. The concatenation-based fusion allows LiDAR/camera features to dominate during gradient updates. The decrease in night scenario (C+L 18.66 → C+L+R 18.45) is likely due to noise-dominated gradients from radar in low-density scenarios where temporal misalignment and pose noise affect sparse radar measurements more severely than dense LiDAR. Radar provides meaningful improvements in rainy scenario (C+L 29.19 → C+L+R 29.86), particularly for dynamic objects, demonstrating its value in adverse weather conditions. It also provides meaningful improvements when the camera + radar configuration is compared to the camera-only model.
>
> We reframed radar’s contribution as providing complementary robustness in specific conditions (rain, dynamic objects) rather than universal performance gains, with explicit acknowledgment of limitations in the conclusion.
>
> Weakness 3 and Question 1:
> We agree that the concatenation-based fusion mechanism has limitations, particularly for integrating sparse radar data, where denser camera and LiDAR features can dominate during training. The novelty of our approach lies in the Gaussian-space fusion paradigm, and the underlying operation can be a simple concatenation or some more complex operation. The concatenation-based approach was chosen deliberately for its computational efficiency and direct integration with our iterative refinement architecture. Even though we are aware that there are other fusion methods for radar data that could be tested (like cross-attention, per-modality gating, or confidence reweighting), we didn’t have enough time to test them.  Weakness 4: We plan on refining the illustrations of the model with a cleaner layout.  Question 2:
> We appreciate the suggestion to validate generalization on SemanticKITTI. Unfortunately, due to computational resource limitations and the tight rebuttal timeline, we cannot provide full SemanticKITTI experimental results at this time. We mentioned it as a possible future work in the conclusion of the revised paper.
>
> Question 3:
> Like many similar papers, we didn’t include visual comparisons with previous methods as we wanted to present our results in as many conditions as possible, while keeping the images in the figures big enough for comparison, and the paper length was limited.
>
> Question 4:
>  Yes, you are correct. The refined Gaussian properties from block N serve as input Gaussians for block N+1, enabling progressive refinement through our Gaussian fusion blocks. The ablation study on the number of refinement blocks was conducted before some final tweaks and tuning, making the results less comparable to other reported results, which is why we didn’t include it. Based on those results, we determined that 4 blocks is the optimal choice regarding the accuracy, inference time, and memory usage. Retraining the model with different numbers of refinement blocks would take too much time with our current resources and wouldn’t fit within our available time.

---

### Author Response · Authors · 2025-12-03
**Overall Summary for Area Chairs: GaussianFusionOcc**

We regret the situation created by the identity leak and appreciate your time in managing this modified review process. To facilitate your evaluation, we provide this summary of substantial improvements made during the review period and our engagement with reviewer concerns.

**Major Improvements During Review Period**

**Strengthened Conceptual Framing**: We rewrote the introduction and method sections to explicitly position Gaussian-space fusion as a paradigm shift from traditional approaches. The key distinction is that fusion occurs on learnable geometric-semantic primitives rather than fixed-resolution grids, enabling sublinear memory growth. Each Gaussian acts as a persistent carrier of multi-modal evidence across iterative refinement, fundamentally different from pixel/voxel-level fusion at static grid locations.

**Nuanced Analysis of Modality Contributions**: We refined our presentation to accurately characterize radar's value proposition. Radar demonstrates meaningful improvements in the majority of scenarios and provides valuable complementary information for dynamic object detection and adverse weather conditions. We reframed the discussion to emphasize radar's role in providing complementary robustness rather than universal performance gains.

**Enhanced Technical Analysis**: We added comprehensive latency profiling showing feature extraction dominates computational cost (76.2%), while GaussianFusionBlocks contribute only 15.6% and splatting 8.1%, validating our representation's efficiency. In our response to the reviewer, we provided mathematical justification for full Gaussian parameterization based on the probabilistic superposition framework from GaussianFormer-2, explaining how opacity enables dynamic confidence encoding during multi-modal fusion.

**Improved Presentation**: We refined the figure illustration of the model and expanded the conclusion to mention future work directions, including evaluation on additional benchmarks and exploration of alternative fusion mechanisms.

**Main Reviewer Concerns and Our Defense**

**Novelty and Conceptual Depth**: Multiple reviewers questioned advancement beyond GaussianFormer. We defended our position emphasizing three contributions:
- the fusion domain paradigm shift, moving from fixed-grid fusion to fusion on adaptive primitives that concentrate modeling capacity where needed,
- the modality-agnostic architecture enabling plug-and-play sensor integration without modality-specific heads,
- the iterative refinement mechanism that allows Gaussians to progressively accumulate multi-modal evidence while simultaneously optimizing geometric configuration and semantic predictions.

**Fusion Design Choices**: We clarified that our core contribution lies in the Gaussian-space fusion paradigm rather than the specific fusion operation. The concatenation+MLP approach was chosen for computational efficiency while achieving state-of-the-art results with superior efficiency. More complex mechanisms could be explored as future work.

**Evaluation Scope**: We explained our strategic decision: nuScenes provides synchronized camera/LiDAR/radar data essential for validating multi-modal fusion (SemanticKITTI lacks radar entirely). The three major benchmarks use identical nuScenes sensor data with different annotation protocols. Given resource constraints, we prioritized thorough evaluation on one protocol with comprehensive ablations over shallow evaluation across multiple protocols.

**Technical Details and Analysis**: Reviewer gF9C's request for a latency breakdown led us to add comprehensive profiling directly in the revised paper. In our response, we addressed concerns about full Gaussian parameterization by referencing prior work demonstrating significant benefits of incorporating opacity, with probabilistic Gaussian superposition providing strong theoretical support.

**Summary**

The revised paper provides substantially clearer conceptual positioning, comprehensive technical analysis, and nuanced discussion of multi-modal contributions. We have thoroughly addressed all reviewer concerns through detailed responses and paper revision.

---

### Meta-Review · Area_Chair_YrG1 · 2026-01-14

**Summary:**

The paper proposes Gaussian FusionOcc, a multi-modal framework for 3D semantic occupancy prediction that integrates Camera, LiDAR, and Radar data using 3D Gaussians. While reviewers acknowledged the efficiency of the Gaussian representation and the validity of the multi-modal approach, the consensus leans toward rejection. The primary concerns stem from limited novelty compared to existing Gaussian-based methods, insufficient experiments (only one benchmark evaluated) , and questionable efficacy of the radar integration, which showed negligible gains or even degradation in some scenarios.

**Reviewer Concerns:**

The rebuttal credibly addresses a subset of clarification level issues. It adds a concrete latency breakdown and argues that feature extraction dominates compute, with smaller shares attributed to the Gaussian fusion blocks and splatting. It also provides a principled justification for evaluating on nuScenes with SurroundOcc, emphasizing synchronized camera LiDAR radar availability and comparability to GaussianFormer style baselines. However, novelty is still perceived as limited by at least one reviewer, who explicitly characterizes the method as an incremental extension with concatenation plus MLP fusion and strong per sensor encoders that confound attribution of gains. The radar contribution concern remains unresolved. Finally, the evaluation breadth concern persists.

**Reviewer Scores:**

Reviewer njS2: (4 to 4). This reviewer maintained concerns about the negligible improvement from radar and the simplicity of the fusion mechanism.

Reviewer bBSw: (4 to 4) After the rebuttal, this reviewer explicitly stated they remained cautious, finding the reliance on a single dataset insufficient for validation and maintaining their score due to limited conceptual depth.

Reviewer iS12: (4 to 4). This reviewer found the contribution below the bar, noting the method is close to prior work and the experiments do not decisively prove that Gaussian-space fusion is the essential factor for the gains.

Reviewer gF9C: (6 to 6). This reviewer was the most positive, appreciating the efficiency and novelty of fusion in Gaussian space. However, they still raised significant concerns about the lack of broader benchmarking and outdated baselines.

---

### Decision · Program_Chairs · 2026-01-26

Reject